# A Novel, Cell-Compatible Hyaluronidase Activity Assay Identifies Dextran Sulfates and Other Sulfated Polymeric Hydrocarbons as Potent Inhibitors for CEMIP

**DOI:** 10.3390/cells14020101

**Published:** 2025-01-11

**Authors:** Anja Schmaus, Sofia Spataro, Paul Sallmann, Stephanie Möller, Leonardo Scapozza, Marco Prunotto, Jonathan P. Sleeman

**Affiliations:** 1European Center for Angioscience (ECAS), Medical Faculty Mannheim, University of Heidelberg, 68167 Mannheim, Germany; anja.schmaus@medma.uni-heidelberg.de (A.S.); paul.sallmann@medma.uni-heidelberg.de (P.S.); 2School of Pharmaceutical Sciences, University of Geneva, Rue Michel-Servet 1, 1211 Geneva, Switzerland; sofia.spataro@hotmail.com (S.S.); leonardo.scapozza@unige.ch (L.S.); marco.prunotto@unige.ch (M.P.); 3Institute of Pharmaceutical Sciences of Western Switzerland, University of Geneva, Rue Michel-Servet 1, 1211 Geneva, Switzerland; 4Biomaterials Department, INNOVENT e.V., Prüssingstrasse 27b, 07745 Jena, Germany; s.moeller@innovent-jena.de; 5Karlsruhe Institute of Technology (KIT) Campus Nord, Institute of Biological and Chemical Systems-Biological Information Processing, 76344 Eggenstein-Leopoldshafen, Germany

**Keywords:** CEMIP, HYBID, hyaluronidase, hyaluronan, dextran sulfate, heparin, polystyrene sulfonate, sulfated hyaluronan

## Abstract

Hyaluronan (HA) levels are dynamically regulated homeostatically through biosynthesis and degradation. HA homeostasis is often perturbed under disease conditions. HA degradation products are thought to contribute to disease pathology. The hyaluronidase CEMIP requires the presence of living cells for its HA depolymerizing activity. CEMIP is overexpressed in a variety of pathological conditions, and the inhibition of its hyaluronidase activity therefore has therapeutic potential. To identify novel inhibitors of the CEMIP hyaluronidase activity, we established here a cell-compatible, medium-throughput assay for CEMIP-dependent HA depolymerization. The assay employs ultrafiltration plates to separate low- from high-molecular-weight HA, followed by quantification of HA fragments using an HA ELISA-like assay. Using this assay, we tested a range of compounds that have been reported to inhibit other hyaluronidases. Thereby, we identified several sulfated hydrocarbon polymers that inhibit CEMIP more potently than other hyaluronidases. One of these is heparin, a sulfated glycosaminoglycan produced by mast cells that constitutes the first described physiological CEMIP inhibitor. The most potent inhibitor (IC_50_ of 1.8 nM) is dextran sulfate, a synthetic sulfated polysaccharide. Heparin and dextran sulfate are used in numerous established and experimental biomedical applications. Their ability to inhibit CEMIP needs to be taken into account in these contexts.

## 1. Introduction

Hyaluronan (HA) is an unbranched glycosaminoglycan consisting of repeating disaccharide units of N-acetyl-D-glucosamine and D-glucuronic acid. Despite its simple structure it is involved in many key molecular and cellular processes. In addition to playing a structural role in matrix organization, it regulates cell signaling, thereby influencing many physiological and pathophysiological processes, including tissue regeneration, wound repair, morphogenesis, and fibrotic diseases. The biological functions of HA are largely dependent on the molecular weight of HA, and high-molecular-weight (HMW) and low-molecular-weight HA (LMW-HA) in part exert opposing functions [1]. Thus, the degradation of HA by enzymes called hyaluronidases is thought to be a decisive factor in determining the function of HA.

In addition to well-established hyaluronidases such as Hyal1, Hyal2, and PH20/Spam, research in recent years has shown that CEMIP (cell migration-inducing and hyaluronan-binding protein), also known as KIAA1199 or HYBID, contributes to the degradation of HA [2,3]. CEMIP is a multidomain protein of 153 kDa that interacts intracellularly with numerous proteins and thereby regulates a variety of signaling pathways. Secreted CEMIP has HA depolymerizing activity, but only in the presence of the cells that secrete it. CEMIP plays a crucial role in the HA metabolism of fibroblasts. Knock-down or knock-out of CEMIP in these cells significantly abolishes HA degradation, leading to an accumulation of HMW-HA [4,5,6]. As fibroblasts are considered to be the main producers of HA in the body, CEMIP expression in these cells may be crucial for the homeostatic regulation of HA size, as well as in pathological conditions where elevated CEMIP levels are frequently observed [3,7,8,9].

Accumulating evidence shows that CEMIP is involved in many different diseases. Examples include diseases of the nervous system, such as hearing loss and multiple sclerosis [10,11]; and inflammatory diseases, such as rheumatoid arthritis and osteoarthritis [7,12,13], as well as inflammatory bowel disease [9]. Most prominently, CEMIP expression in numerous types of cancer has been functionally linked to tumor progression, metastasis, and therapy resistance [3]. Studies that have investigated the molecular mechanisms through which CEMIP contributes to these diseases rarely address the hyaluronidase activity of CEMIP, even though this might be highly relevant due to disease-relevant perturbation of HA metabolism. This is in part due to the fact that there are no well-established methods to investigate the hyaluronidase activity of CEMIP, and also due to the lack of specific inhibitors for the hyaluronidase function of CEMIP that can discriminate between HA/hyaluronidase dependent and HA/hyaluronidase independent functions. Given the plethora of diseases in which CEMIP has been implicated, potent inhibitors for the hyaluronidase activity of CEMIP would also represent promising therapeutic tools.

The hyaluronidase activity of CEMIP contrasts in several respects to that of other hyaluronidases. In addition to the dependency of this activity on the presence of the cells that secrete it, the structure of CEMIP also has no homology with other known hyaluronidases apart from TMEM2. HA depolymerization by CEMIP is thought to require secretion of the protein, followed by internalization of the protein together with HA in a clathrin-dependent manner, and then internalization into endosomes. It is not known exactly where and when during this process HA degradation takes place, but the HA degradation products are subsequently found extracellularly [4,14,15].

To date, the assays that have been used to analyze the hyaluronidase activity of CEMIP involve the addition of HMW-HA (either ^3^H-HA or fluorescently labeled HA) to cells in culture (either fibroblasts expressing endogenously high levels of CEMIP or 293 cells overexpressing CEMIP), followed by analysis of the HA degradation products using semi-quantitative analytical methods such as size-exclusion chromatography [4,6,16] or agarose gel electrophoresis [5,17]. Although these methods have significantly improved our understanding of CEMIP’s hyaluronidase function, they have several limitations. The use of radioactively labeled HA limits its broad application due to safety and regulatory issues. The labeling of HA with fluorochromes can potentially influence the binding characteristics of CEMIP to the HA, and the fluorochrome labeling is also often not stable, making it difficult to compare experimental results [18]. Size-exclusion chromatography is time consuming, and quantification is laborious. None of the methods currently employed are suitable for medium- to high-throughput chemical library screens.

Previously we have employed cell-based assays coupled to agarose gel electrophoresis in order to identify sulfated HA as a potent CEMIP inhibitor [5]. However, this assay is unsuited for the screening of a large number of test substances in a timely or quantitative manner. To support the identification of other CEMIP hyaluronidase activity inhibitors, the aim of this study was therefore to develop a robust assay that allows the simultaneous testing of potential inhibitors of the HA degrading activity of CEMIP in a medium-throughput format. The principle of the assay is to grow CEMIP-secreting cells in 96-well plates in the presence of HMW-HA. The intermediate-molecular-weight (IMW) HA and LMW-HA produced by the CEMIP hyaluronidase activity is then separated from undegraded HMW-HA using 96-well ultrafiltration plates. Levels of IMW- and LMW-HA in the ultrafiltrates are then quantified using an HA ELISA-like 96-well assay. Concomitantly, the toxicity on the CEMIP-producing cells of the compounds under investigation is tested to exclude false-positive hits. As a proof of principle, we used this assay to assess the ability of a panel of compounds that have been previously shown to inhibit the activity of other hyaluronidases for their ability to inhibit the HA degrading activity of CEMIP. Thereby, we found that a number of sulfated polymeric hydrocarbons are novel, potent hyaluronidase inhibitors of CEMIP. Dextran sulfate emerged from these studies as the most potent inhibitor of the CEMIP hyaluronidase activity that has been described to date.

## 2. Materials and Methods

### 2.1. Cell Culture

293T cells were cultivated in DMEM, 10% FCS, 1% penicillin–streptomycin. Primary mouse embryonic fibroblasts (MEFs) were isolated from E13.5 embryos of mice from a mixed background (C57Bl/6 FVB) and cultivated in DMEM, 10% heat-inactivated FCS, and 1% penicillin–streptomycin as previously described [5,19]. MRC5 cells (human fetal lung fibroblasts) were a kind gift from Thordur Oskarsson, DKFZ and were cultivated in MEM, 10% FCS, 2 mM glutamine, 1% non-essential amino acids, and 1% penicillin–streptomycin. All cells were cultivated at 37 °C, 5% CO_2,_ 21% O_2_.

### 2.2. Plasmids and Transfection

The 293T cells stably expressing human or mouse CEMIP have been previously described [5]. The plasmids containing the murine or human cDNA sequence of CEMIP in the pRP vector were purchased from VectorBuilder (vector ID murine CEMIP: VB180316-1177crn; vector ID human CEMIP: VB191004-1084jkc; vector ID empty vector control: VB171110-1101tcx). Cells were transfected with CEMIP expression vectors or controls using Lipofectamine 2000, and stable clones were selected with puromycin. CEMIP expression and activity were analyzed by Western blot and hyaluronidase activity assays, and clones with the strongest expression were selected.

### 2.3. Chemicals

Delcore (HA chemically modified by oleic acid) was purchased from Contipro (specification 600-19-01). Ipriflavone (Cat # 16499), heparin (Cat #: H3149), poly(4-styrenesulfonic acid) sodium salt (PSS) 1000 and 70 (Cat #: 434574 and 243051), glycyrrhizate ammonium salt (Cat #: G0460000), N-acetyl-L-cysteine (Cat #: A9165), and ascorbic acid 6-palmitate (Vcpal, Cat #: A1968) were all obtained from Merck. Glutathione (Cat #: 6832), dextran sulfate 500 (DXS 500, Cat #: 5956), dextran sulfate 40 (DXS 40, Cat #: 7612), dextran sulfate 8 (DXS 8, Cat #: 7610), dextran 500 (Cat #: 9219), and dextran 40 (Cat #: 7626) were ordered from Roth. Ipriflavone, glycyrrhizate ammonium salt, and Vcpal were dissolved in 10% DMSO. All other substances were dissolved in H_2_O. Additional standard chemicals were obtained from Merck or Roth.

### 2.4. Sulfated Hyaluronan

The sulfated HA derivatives sHA1.2 (36), sHA2.0 (18), sHA2.5 (7), sHA3.5 (21), sHA3.6 (29), and sHA3.7 (108) were synthesized by INNOVENT e.V. using HMW-HA obtained from either Aqua Biochem or from Kraeber. The synthesis of sulfated GAG derivatives has been described previously [5,20,21]. The respective degree of sulfation (Ds) and the molecular weight were also determined as recently described [5]. High- and low-sulfated HA with a sulfation degree of 13.9 or 9.7% respectively (corresponding to a Ds of approximately 3.5 or 2.0) were purchased from TCI chemicals (Tokyo, Japan, Cat #: H1740 and H1739).

### 2.5. CEMIP Hyaluronidase Activity Assay

The CEMIP hyaluronidase activity was assessed using cells in culture, as described previously [5]. Assays were performed in 24- or 96-well plates in a total volume of 500 or 100 µL medium, respectively. In general, cells were seeded at concentrations such that they were confluent at the end of the assay. 293T cells expressing mouse or human CEMIP or empty vector controls were seeded in 24-well plates at 240,000 cells/well for 24 h incubations with HA. In 96-well plates, if not otherwise stated, 40,000 cells/well were seeded. For a 72 h assay duration (Figure 1), 293T cells were seeded at 80,000 cells/well, and MRC5 cells and MEFs were seeded at 50,000 cells/well in 24-well plates. The cells were cultivated for 24 h, and then inhibitors (from 20× stock solutions) or the appropriate controls (H_2_O or 10% DMSO, see above), and HMW-HA (1.5 MDa, Lifecore Biomedical, Menlo Park, CA, USA, Cat # HA 15M-1) were added. The HA was prediluted in medium at 500 µg/mL and then carefully pipetted to the cells to reach a concentration of 50 µg/mL HA, or as indicated. Cells were then further cultivated with HA at 37 °C for 24 h, unless otherwise stated. HA samples were then analyzed by agarose gel electrophoresis or by means of ultrafiltration and HA ELISA-like assay, as described below.

### 2.6. Analysis of HA Degradation Products by Agarose Gel Electrophoresis

For analysis of samples from CEMIP hyaluronidase activity assays by agarose gel electrophoresis, 100 µL culture medium was collected (containing 5 µg HA) and treated with 0.1 mg/mL Proteinase K, 0.01% SDS for 4 h at 60 °C. HA was precipitated with 4 volumes of absolute ethanol by incubation at −20 °C overnight. Samples were then centrifuged (10,000× *g* for 10 min); pellets were washed with 70% ethanol; and, after air-drying at room temperature, were resuspended in H_2_O.

Electrophoresis of HA is based on the method of Lee and Cowman [22] and was performed according to a protocol provided by Cleveland Clinic (NHLBI award number PO1HL107147). HA samples were mixed with loading buffer (2M sucrose + 0.01% w/v bromophenol blue sodium salt) and separated by gel electrophoresis using 1% agarose (Biozym Scientific, Hessisch Oldendorf, Germany, Cat # 840004) in TAE buffer (40 mM Tris, 20 mM glacial acetic acid, 1 mM EDTA, pH 8.0). After the run, the gel was equilibrated with 30% ethanol for 1 h on a shaker and then stained overnight, in the dark, with 2.5 mg/mL Stains-All (Merck, Darmstadt, Germany, Cat # E9379) in 30% ethanol. For destaining, the gel was incubated for several hours in the dark in H_2_O, then exposed to light until the background was sufficiently reduced, and subsequently photographed. Undigested HMW-HA (1.5 MDa) and 50 kDa HA (Select-HA from Echelon Biosciences, Salt Lake City, UT, USA, Cat # HYA-0050) served as size standards. As additional control HA (<10 kDa) was prepared from partially digested HA using bovine testis hyaluronidase (Merck Cat #: H3884), which was then centrifuged through 10 kDa Amicon filters (Millipore, Darmstadt, Germany, Cat # UFC 501096).

For quantitative analysis of HA agarose gels stained with Stains-All and to calculate IC_50_ values, ImageJ (1.54k 15 September 2024) [23] was used to assess intensities of HA levels in the gels, as described previously [5]. First, a color deconvolution filter (Fast Red Fast Blue DAB) was applied to the images to reduce unspecific signals, for example, the brown bands derived from heparin or dextran sulfates on the gels. From the image with the strongest signal, a negative image was created using the tool “invert”. By using the line selection tool with an appropriate line width, plot profiles were measured in each lane, displaying the intensities of pixels along the individual lanes. To reduce the background signal, the minimum grey value of the image was subtracted from the resulting intensities. The gravity of the plot profile curves was then calculated at 50% of the integrated density of each lane. For normalization, HA without inhibitor was set to 0%, and undigested HA was set to 100% inhibition. The resulting values were used for calculation of a four-parameter, sigmoidal dose–response curve using non-linear regression and assessment of IC_50_ values with GraphPad Prism (version 10).

### 2.7. Analysis of CEMIP HA Degradation Products by Ultrafiltration and HA-ELISA

For quantitative analysis of samples from CEMIP hyaluronidase activity assays, assays were performed in 96-well plates with 40,000 cells/well. After 24 h incubation with HA, the cell culture supernatant was transferred to new 96-well plates and then centrifuged for 5 min at 400× *g* to remove cell debris. Then, 50 µL of the supernatant was diluted with 50 µL diluent from the HA ELISA-like assay (Echelon Biosciences, Cat #: K-1200) and centrifuged through Acroprep Advance 100 kDa Omega 96-well plates (Pall, Cortland, NY, USA, Cat #: 8036) at 1500× *g* for 10 min at 4 °C. HA concentrations in the ultrafiltrates were then analyzed at a dilution of 1:10 or 1:20 using an HA ELISA-like assay (Echelon Biosciences, Cat #: K-1200) according to the manufacturer’s protocols. A four-parameter dose–response curve using non-linear regression was generated from the standards with GraphPad Prism, which was used to calculate sample values.

### 2.8. Western Blot

Cells were lysed in Laemmli buffer (125 mM Tris HCl pH 6.8, 4% SDS, 20% glycerol) and total protein concentrations were measured using BCA assays. To analyze secreted CEMIP, conditioned medium was collected, spun down at 500 g 5 min and mixed with Laemmli buffer. After the addition of 5% β-mercaptoethanol and bromophenol blue, the samples were heated for 5 min at 95 °C, and then subjected to SDS-PAGE. Proteins in the gels were transferred to Immobilon-P PVDF membrane (Merck Millipore, St. Louis MI, USA) using standard Western blotting techniques. The membranes were probed with polyclonal anti-CEMIP (1 µg/mL, ARP42526_P050, Avivasystems, San Diego, CA, USA) or anti-α-tubulin (0.2 µg/mL, T8203, Merck, Boston, MA, USA) antibodies. HRP-conjugated secondary antibodies were from DAKO. The protein bands were visualized using the Pierce ECL or ECL Dura Western Blotting Substrate (Thermo Fisher Scientific, Waltham, MA, USA).

### 2.9. CPC Pulldown

To assess the binding ability of CEMIP to HA, cetylpyridinium chloride (CPC) pulldown experiments were performed similar to previously published protocols [24,25]. 135 μL of protein solution (0.25–1 µg/µL) was mixed with 50 μL of 1 mg/mL HA (50 kDa, 100 kDa or 1.5 MDa, Lifecore Biomedical) or water, and incubated for 1 h at 37 °C. After adding 460 μL of 1.4% CPC aqueous solution (Sigma, Taufkirchen, Germany), the samples were incubated again for 1 h at 37 °C to precipitate the HA. Samples were then centrifuged for 10 min at 13,000× *g* and the precipitates were washed 3 times with 1 mL 30 mM NaCl, 1% CPC. Pellets were dissolved in 50 μL Laemmli buffer containing DTT and heated at 95 °C for 10 min. Samples were analyzed by Western blot using antibodies specific for CEMIP as described above.

### 2.10. Proliferation Assays

The CyQuant Direct Cell proliferation assay Kit (Thermo Fisher Scientific, Waltham, MA, USA, Cat #: C35011) was used to assess viable cell numbers by means of a cell-permeable DNA-binding dye. The fluorescent dye is used in combination with a masking dye reagent, which blocks the staining of dead cells or cells with compromised cell membranes. Cells were seeded into 96-well plates, treated and cultivated as described above for hyaluronidase activity assays. CyQuant Direct detection reagent was prepared by mixing the nucleic acid stain (0.2%) and the background suppressor (1%) in HBSS. After cultivation, cells were incubated with 50 μL CyQuant Direct detection reagent at 37 °C for 30 min. The fluorescence intensity of each well was measured from the bottom using a SpectraMax iD3 microplate reader (Molecular Devices, San Jose, CA, USA) with excitation at 480 nm and emission at 530 nm.

### 2.11. Statistical Analysis

Comparisons between different samples were performed with GraphPadPrism software (version 10) using one-way ANOVA. Statistical significance was set at *p* < 0.05. Data are expressed as mean +/− SE. (* *p* < 0.05, ** *p* < 0.005, *** *p* < 0.001, **** *p* < 0.0001).

## 3. Results

### 3.1. Establishment of a Medium-Throughput Assay for CEMIP-Mediated HA Degradation

The design of the medium-throughput cell-based HA degradation assay for CEMIP is illustrated in Figure 1A. CEMIP-expressing cells are grown in a 96-well format, and incubated together with HWM-HA (circa 1.5 MDa (approximately 3750 disaccharides in length)) in the presence or absence of substances to be screened for possible inhibition of the CEMIP hyaluronidase activity. The size of 1.5 MDa HA was chosen as cells usually produce HA in this HMW range [26]. Furthermore, the use of 1.5 MDa HWM HA as a CEMIP substrate facilitates separation of undigested HA from the HA fragments using ultrafiltration plates. The concentration of 50 µg/mL is physiologically relevant (for example synovial fluid HA concentrations reach several mg/mL [27]). Moreover, the use of a concentration of 50 µg/mL reduces variance in the assay caused by any endogenous production of HA by the cells. After an appropriate cultivation period, the conditioned medium is harvested and transferred to 96-well ultrafiltration plates with a molecular weight cut-off of 100 kDa (circa 250 disaccharides). The filtrates are then tested for HA content in an ELISA-like HA assay that detects HA fragments larger than 6 disaccharides in length (2.4 kDa) [28]. This allows HA fragments of between 6–250 disaccharides in length (2.4–100 kDa) produced by the CEMIP hyaluronidase activity to be quantified. In parallel, the impact of the test substance on cell numbers in the original culture plate is assessed, to exclude false-positive test substances that impact on cell proliferation or exert a toxic effect, which would result in the detection of reduced HA fragmentation but not due to direct inhibition of the CEMIP hyaluronidase activity.

**Figure 1 cells-14-00101-f001:**
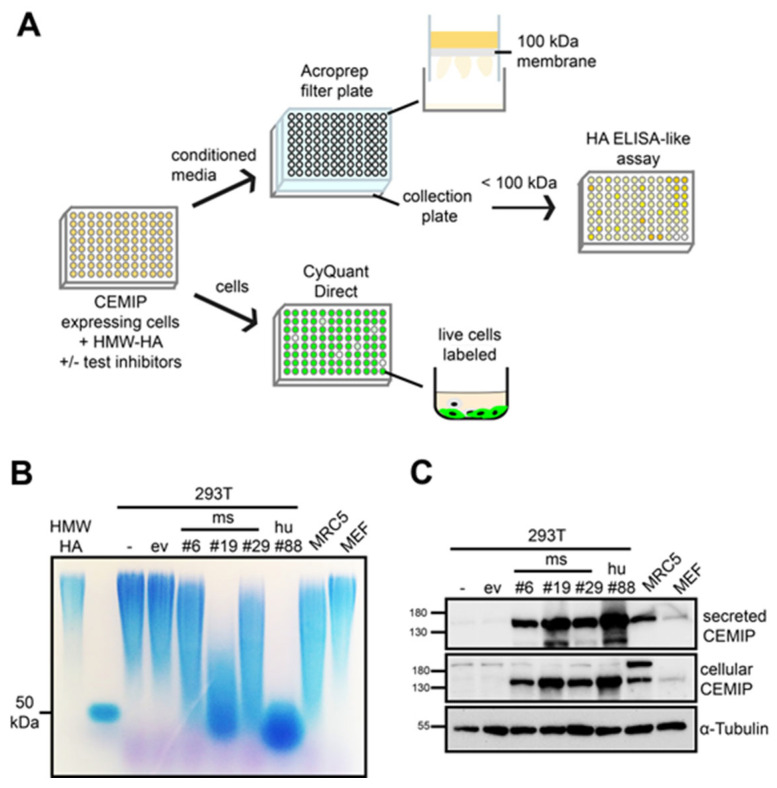
(**A**) Schematic diagram of the medium-throughput assay. (**B**,**C**) CEMIP expression correlates with the degree of HA degradation. Stable clones from 293T cells transfected with mouse or human CEMIP expression plasmids (ms CEMIP clone #6, #19, #29 or hu CEMIP clone #88) were analyzed by hyaluronidase activity assay (**B**) and Western blot (**C**). Untransfected cells (−) or cells transfected with empty vector (ev) served as controls. In addition, MRC5 fibroblasts and MEFs were analyzed. All cells were treated for 72 h with 1.5 MDa HMW-HA. (**B**) For the hyaluronidase activity assay, the conditioned medium was harvested, treated with Proteinase K, then HA was precipitated with EtOH. The precipitated HA was analyzed by 1% agarose gel electrophoresis and stained with Stains-All. HMW-HA (1.5 MDa) which was incubated in medium only, and HA of a defined size (50 kDa) served as size standards. (**C**) Lysates of the cells or an aliquot of the conditioned media were analyzed by Western blot to assess the amounts of cellular and secreted CEMIP. Probing the blots with anti-α-tubulin antibodies served as loading control for the lysates.

To establish this assay, we first identified the most suitable cell line for use in the medium-throughput CEMIP hyaluronidase activity assay. To this end, we compared the HA degrading activity of different cell lines with ectopic or endogenous CEMIP expression. Initially, we analyzed HA degradation by agarose gel electrophoresis, as this method allows partial degradation of HA to be assessed. As shown in Figure 1B,C, the expression of CEMIP intracellularly as well as the amount of secreted CEMIP correlates directly with the HA degrading activity observed in the agarose gel electrophoresis assays. No degradation was observed in wild-type 293T cells, or wild-type 293T cells transfected with empty vector, excluding the possibility that endogenous expression of hyaluronidases by the CEMIP-expressing 293T cells contributes to the hyaluronidase activity observed. Furthermore, the 293T cells produced no detectable HA endogenously.

The strongest HA degrading activity was observed in single cell clones of 293T cells that ectopically express mouse or human CEMIP (clone #19 and #88, respectively) [5]. For comparison purposes, we also analyzed MRC5 and MEF, two different fibroblast lines that have endogenous CEMIP expression. MRC5 fibroblasts express CEMIP and degrade HA to a similar level as 293T cell clones with medium levels of ectopic CEMIP expression (clones #6 and #29), indicating that the ectopic expression of CEMIP in the 293T cell clones is within a physiological range. Similar to MRC5 cells, MEFs regulate their endogenous HA turnover via CEMIP [5], but express relatively low CEMIP levels which were not sufficient to observe HA degradation in the experimental settings employed here, probably due to the relatively high HA concentrations used in the assay. MRC5 and MEFs were found to produce around 1 μg/mL HA endogenously under the conditions employed, indicating that endogenous HA production was not a confounding factor in the assay. Based on these results, we concentrated on the 293T cell clones that ectopically express the highest levels of mouse or human CEMIP (ms clone #19 and hu clone #88) for further tests of the hyaluronidase activity of CEMIP and the establishment of the medium-throughput CEMIP hyaluronidase activity assay.

Agarose gel electrophoresis allows the analysis of HA fragments over a broad molecular weight range of HA. In contrast, the ultrafiltration filters to be used in the medium-throughput assay have a 100 kDa cut-off, which means that any HA that is partially degraded by CEMIP but which has a molecular weight above 100 kDa would not be detected in the assay. To ensure that all CEMIP HA-degrading activity is detectable in the assay, these constraints demand that in the absence of inhibitor, the size of CEMIP-generated HA fragments should be below 100 kDa in size. The 72 h digest shown in Figure 1 produces HA fragments of around 50 kDa, which would in principle provide a satisfactory baseline for CEMIP-mediated HA degradation in the absence of inhibitor. However, a 72 h digestion period would increase the risk that non-specific effects of potential CEMIP inhibitors under test may emerge due to the inhibition of cell proliferation or toxicity. We therefore next established the optimum number of cells and incubation time required to achieve complete degradation of HWM-HA into HA fragments smaller than 100 kDa in size, in as short a time as possible.

Initial tests showed that the extent of HA degradation correlates with the number of cells used in the assay. HA of around 50 kDa was produced after 24 h only by confluent cells, corresponding to more than 40,000 cells/well (Figure 2A,B). Under confluent conditions, partial HA degradation was already visible after 1 h but continued out to 24 h, at which time HA fragments of around 50 kDa accumulated (Figure 2C).

Hyaluronidases such as Hyal1 or bovine testis hyaluronidase (BTH) completely degrade HA into fragments of 2–3 disaccharides [29], which are not detectable using agarose gel electrophoresis or the ELISA-like HA assay. By contrast, the smallest HA fragments produced by CEMIP have been suggested to be 8 disaccharides (human CEMIP) or 10 disaccharides (mouse CEMIP), which although only 3–4 kDa in size are large enough to be detected by the ELISA-like HA assay. However, the production of even smaller HA fragments by CEMIP has also been suggested [14], which conceivably may not be detectable in the ELISA-like HA assay, and which would potentially give a non-quantitative readout of CEMIP activity in the medium-throughput assay. Under the conditions employed here, the limit of CEMIP-mediated HA fragmentation is slightly below 50 kDa (Figure 1 and Figure 2). Nevertheless, as CEMIP can bind to HA of 50 kDa (Figure 2D) and could therefore in principle further degrade HA of this size into smaller fragments, we next set out to exclude the possibility that CEMIP produces significant amounts of small HA fragments that would not be detected in our medium-throughput assay. To this end, we compared HMW-HA and 50 kDa HA as substrates for CEMIP. Only human but not mouse CEMIP was able to marginally degrade HA of 50 kDa further, and the size of the resulting HA degradation products were comparable in size to those produced by CEMIP-degraded HMW-HA (Figure 2E). To exclude the possibility that CEMIP activity might diminish over time in culture, we also took conditioned medium containing HMW-HA HA degraded by CEMIP-expressing cells and added it to fresh CEMIP-expressing cells. We then determined whether the HA fragments were further reduced in size by the fresh cells but found only a marginal further degradation (Figure 2F). Taken together, these results suggest that under the conditions employed in the medium-throughput assay, virtually all the HA degraded by the CEMIP-expressing cells will be between 10–50 kDa in size (Figure 2F), and therefore quantitative assessment of CEMIP-mediated HA degradation can be achieved with the assay.

### 3.2. The Medium-Throughput Assay for CEMIP-Mediated HA Degradation Gives Equivalent Results to Those Obtained with Agarose Gel Electrophoresis

As a proof of principle for our medium-throughput assay, we used sulfated HA (sHA), a potent inhibitor of CEMIP that we have previously shown has an IC_50_ for CEMIP of around 10 nM [5]. Mechanistically, we found in a CPC pulldown assay that highly sulfated HA potently prevents the binding of HA to CEMIP (Figure 3A). We then treated CEMIP-expressing cells with sulfated HA, and compared the inhibition of CEMIP-mediated HA degradation measured by agarose gel electrophoresis with that measured in the medium-throughput assay (Figure 3B,C). CEMIP-mediated degradation of HA and its inhibition by sulfated HA was equivalent in the two assays, with sHA3.7 significantly reducing the degree of HA degradation by about 50% at the IC_50_ concentration of sHA3.7 in the medium-throughput assay, demonstrating the accuracy and sensitivity of the assay (Figure 3C). Furthermore, although we found that the ultrafiltration plates were unable to completely recover the CEMIP-degraded HA fragments during the separation of these fragments from the undegraded HMW-HA, the size distribution and relative abundance of the HA fragments in the filtrates was comparable to that observed in gel electrophoresis assays (Figure 3B,C).

### 3.3. Characterization of Parameters Required for Inhibition of CEMIP-Mediated HA Degradation by Sulfated HA

To test and validate the medium-throughput assay further, we analyzed sulfated HA derivatives of different sizes, degree of sulfation and from different sources, and compared their ability and potency to inhibit CEMIP, extending our previous study in which we assessed the inhibitory effect of sHA using agarose gel electrophoresis [5]. To capitalize on the quantitative measurements afforded by the medium-throughput assay, we extended our previous analyses by treating CEMIP-expressing cells with increasing concentrations of different variants of sulfated HA ranging from 7 to 108 kDa and a degree of sulfation (Ds) from 1.2 to 3.7. As expected from our previous findings, highly sulfated HA (Ds 3.7) of 108 kDa already completely inhibited HA degradation at a concentration of 0.1 µM (Figure 4A). Sulfated HA with a similar degree of sulfation but smaller in size (Ds 3.5, 21 kDa) showed a similar inhibitory efficacy. Interestingly, sulfated HA with a lower sulfation degree of 2.0 also showed a similar potency. By contrast, weakly sulfated HA (Ds 1.2, 36 kDa) and a very small sulfated HA derivative of only 7 kDa but with a Ds of 2.5 showed a weaker, but nevertheless concentration dependent inhibitory effect (Figure 4A).

Next, we analyzed the efficacy of inhibition of CEMIP-mediated HA degradation by low sulfated HA (Ds 2.0) and high sulfated HA (Ds 3.5) obtained from a commercial source (Figure 4B). As the molecular weight of these substances was not provided by the manufacturer, we estimated their size in comparison to other sulfated HA derivatives of known size on agarose gels (Appendix A). This showed that both the low and highly sulfated HA from the commercial source have a rather high molecular weight of more than 100 kDa in size. We therefore tested the inhibitory effects of these derivatives on CEMIP-mediated HA degradation at concentrations of 1, 10 and 100 µg/mL. For comparison, a 1 µM solution of sHA3.7 with a molecular weight of 108 kDa equates to approximately 100 µg/mL. The results showed that the commercially-available low and high sulfated HA both exhibited a potent inhibition of CEMIP-mediated HA degradation, even with the lowest concentration employed (Figure 4B).

Taken together, these results show that sulfated HA derivatives with a relatively broad range of degree of sulfation (Ds > 2.0) and molecular weight (>18 kDa) are able to inhibit CEMIP-mediated HA degradation to a similar extent, with a trend towards more potent inhibition with a higher degree of sulfation and a larger molecular weight. Importantly, analysis of cell numbers using the CyQuant Direct assay in all these experiments showed no significant decrease in cell numbers, indicating that the substances tested did not affect cell proliferation or exert a toxic effect (Figure 4C,D). Although HA modified with oleic acid (Delcore) partially inhibited CEMIP-mediated HA degradation, this was not in a dose-dependent manner (Figure 4A). As we have previously shown in other assays that Delcore is unable to inhibit the CEMIP hyaluronidase activity [5], we conclude that the sulfation rather than HA modification per se is crucial for the inhibitory effects of sulfated HA.

### 3.4. Screening of a Panel of Hyaluronidase Inhibitors Identifies Dextran Sulfate as a Highly Potent Inhibitor of CEMIP-Mediated HA Degradation

Next, we investigated the ability of a panel of substances that have been reported to inhibit the hyaluronidase activity of Hyal1 and/or testicular hyaluronidase for their ability to inhibit CEMIP-mediated HA degradation. Among them were modified carbohydrates such as heparin, a naturally sulfated glycosaminoglycan [30]; dextran sulfate (DXS), a synthetic, sulfated polysaccharide composed of branched polymers of anhydroglucose [31]; and polystyrene sulfonate (PSS), a synthetic, sulfated polymer made from monomers of the aromatic hydrocarbon styrene [30]. In addition, we tested the synthetic antioxidant N-acetyl-L-cysteine (NAC), a derivative of cysteine [32]; the naturally occurring antioxidant glutathione [32]; glycyrrhizate, a plant-derived saponin [30,33]; and ascorbic acid 6-palmitate (VcPal), an ester formed from ascorbic acid and palmitic acid which is used as antioxidant food additive [34,35]. In addition to these Hyal1 and/or testicular hyaluronidase inhibitors, the panel also included ipriflavone, an isoflavone that has been suggested to influence HA metabolism in CEMIP-expressing chondrosarcoma cells [36].

The ability of this panel of substances to inhibit HA degradation by CEMIP-expressing cells at 0.1, 1, and 10 µM concentrations was assessed using the medium-throughput assay. Heparin inhibited the hyaluronidase activity of CEMIP in a concentration dependent manner, while DXS and PSS completely blocked HA degradation at the concentrations employed. None of the other compounds inhibited the CEMIP hyaluronidase activity at the concentrations of up to 10 µM tested in the assay (Figure 4A). The CyQuant Direct assays used to exclude cytotoxic or anti-proliferative effects of the substances tested revealed that the highest concentrations of PSS clearly compromised the cell viability and/or proliferation, indicating that the reduced HA degradation of these high PSS concentrations might be due at least in part to cytotoxic effects rather than to inhibition of the CEMIP hyaluronidase activity (Figure 4C).

In further experiments, we analyzed the CEMIP inhibitory activity of heparin, DXS, and lower concentrations of PSS more in depth, as all three of these substances are sulfated hydrocarbon polymers, similar to sHA. Titration experiments and analysis thereof by agarose gel electrophoresis revealed that heparin inhibits CEMIP with an IC_50_ of 0.3 µM (Figure 5A,B). Even the highest concentrations of heparin employed had no impact on the number of viable cells (Appendix A), indicating that this inhibitory effect is not due to toxic effects on the CEMIP-expressing cells. Heparin also did not reduce expression of CEMIP intracellularly, and it did not reduce secretion of the protein (Appendix A), ruling out reduced expression of CEMIP as an explanation for the inhibitory effect of heparin.

For PSS, we also evaluated its influence on cell numbers and viability by CyQuant assays in parallel to hyaluronidase activity assays and included not only the high-molecular-weight PSS 1000 (1000 kDa) in the assays, but also the lower-molecular-weight PSS 70 (70 kDa) (Figure 5C–E). The results obtained indicate that the calculated IC_50_ concentrations of 0.5 nM for PSS 1000 and 17.7 nM for PSS 70 should be considered with caution, as reduced cell numbers at higher concentrations of the substances clearly contribute to the apparent inhibition of the CEMIP hyaluronidase activity at these concentrations. Nevertheless, the strong inhibition of CEMIP-mediated HA degradation by PSS 70 and PSS 1000 at concentrations that do not significantly impact on cell viability indicate that the substances are able to specifically inhibit the CEMIP-mediated hyaluronidase activity at low concentrations, indicating that PSS is a potent inhibitor of CEMIP hyaluronidase activity. The high-molecular-weight PSS is a more potent inhibitor than the lower-molecular-weight polymer (Figure 5D,E).

DXS 500 with a molecular weight of 500 kDa completely inhibited CEMIP-mediated HA degradation at 0.1 µM (Figure 4A). In further experiments, we determined that DXS 500 inhibits CEMIP very potently, with an IC_50_ of 1.8 nM (Figure 5F,G), and does not reduce cell numbers (Appendix A) or lower the expression or secretion of the CEMIP protein (Appendix A), demonstrating the specificity of this inhibitory effect. We also analyzed the inhibitory activity of DXS with a smaller size (DXS 40 and DXS 8 with molecular weights of 40 and 8 kDa, respectively), but with a similar sulfation degree of around 20%. DXS 40 and DXS 8 were also effective inhibitors of the CEMIP hyaluronidase activity, but with significantly higher IC_50_ concentrations (~22.2 nM and 1.0 µM, respectively) than DXS 500 (Figure 5F,G). This indicates that the inhibitory activity of DXS correlates with the size of the polymer. Furthermore, non-sulfated Dextran 500 and Dextran 40 had no impact on CEMIP-mediated HA degradation even at high concentrations (Figure 5H), indicating that the inhibitory effect is dependent on the sulfation of DXS.

## 4. Discussion

In this study, we have developed a cell-based assay for the CEMIP hyaluronidase activity in a 96-well format that is suited for medium-throughput chemical library screens to identify CEMIP inhibitors. Using this assay, we have identified heparin, PSS, and dextran sulfate as novel and potent inhibitors of the CEMIP hyaluronidase activity. Together with our previous publication showing that sulfated HA is also a potent inhibitor of CEMIP-mediated HA depolymerization [5], these data highlight sulfated hydrocarbon polymers as a class of compounds that target the CEMIP hyaluronidase activity. The results also identify heparin as the first physiological inhibitor of CEMIP to be discovered and document dextran sulfate as the most highly potent inhibitor of the CEMIP hyaluronidase activity that has been described to date.

HA degradation by CEMIP has, until now, been analyzed by time-consuming and low-throughput assays, such as size-exclusion chromatography and agarose gel electrophoresis [14,16,17], because CEMIP only exhibits hyaluronidase activity in the presence of vital cells that secrete it. For these reasons, many hyaluronidase activity assays [32,34,37,38,39] are unsuited for assaying the CEMIP-mediated HA depolymerization. Although assays using HA-coated plates have been used to identify various hyaluronidase inhibitors [30,40] and could, in principle, be employed to assess the CEMIP hyaluronidase activity, our preliminary experiments revealed that CEMIP-expressing cells adhere poorly to the HA-coated plates, precluding this approach. Recently developed FRET-based techniques for hyaluronidase detection that employ cationic carbon dots [41] could, in principle, be used to assess the CEMIP hyaluronidase activity. However, in preliminary experiments, we found that cationic carbon dots exert considerable toxicity on the CEMIP-expressing cells, making the approach unsuitable. The assay we describe here overcomes these throughput and toxicological limitations. We and others have previously used individual ultrafiltration units (for example, Amicon filters) coupled to an HA ELISA-like assay to detect IMW-/LMW-HA in biological samples [12,28,42,43]. By upscaling this approach to a 96-well format, the CEMIP hyaluronidase activity assay we describe here has considerable advantages compared to other activity assays and facilities medium-throughput analyses.

Our assay employs natural, unmodified HMW-HA as a substrate. In principle, a fluorescently labeled HA (FA-HA) could be used as a substrate instead, allowing the HA ELISA-like assay to be substituted by fluorescence measurements of the ultrafiltrates. However, FA-HA are heterogenous in both HA size and the degree of substitution with fluorescein moieties [18]. Furthermore, substitution with fluorescent moieties would be likely to affect the ability of CEMIP to bind to and/or cleave the fluorescently-modified HA, as has recently been observed for TMEM2 [18]. In this regard, we note from our own work that substitution of HA with sulfate groups potently inhibits the CEMIP hyaluronidase activity. The size of cleavage products of FA-HA and the possible presence of free fluorescent moieties may also impact on the specific detection of degraded HA in the ultrafiltrates.

A limitation of the CEMIP hyaluronidase activity assay we describe here is inaccuracy in the quantitation of very low levels of CEMIP-dependent HA degradation. he Echelon HA ELISA-like assay, one component of our assay, has by itself an intra-assay coefficient of variation (CV) of 18.9% and an inter-assay CV of 9.5% [44]. After calculating the intra-assay CV for our cell-based CEMIP hyaluronidase assay as a whole, we found a strong dependence on the quantity of HA fragments produced. For quantities of CEMIP-generated fragments of HA of 1.3 μg/mL and above in the filtrates, the CV was 12.7% (sixteen triplicate samples). The CV increases progressively when CEMIP activity is reduced and lower amounts of HA fragments are produced, dropping to 32% for all conditions measured (52 triplicate samples). Similarly, the inter-assay CV was 9% for quantities of HA fragments of 1.3 μg/mL and above, dropping to 54% for samples containing less than 0.2 mg/mL HA. These data reflect the fact that competitive ELISA-like assays (such as the Echelon HA ELISA-like assay) are very sensitive to variance in samples with low concentrations of analyte, and the fact that cell-based assays generally have a higher CV compared to non-cell-based assays [45]. The high intra- and inter-assay CVs in our assay under conditions in which CEMIP hyaluronidase assay is absent or strongly inhibited—and, thus, little if any HA fragmentation occurs—means that our CEMIP hyaluronidase assay is not suited for accurate quantification of very low levels of CEMIP activity. Nevertheless, this limitation does not impact on the application of the CEMIP hyaluronidase activity for screening for inhibitors, as employed in our study.

The results we report here indicate that cells expressing high ectopic levels of CEMIP produce HA of approximately 10–50 kDa as the smallest degradation products. Consistently, MRC5 fibroblasts express CEMIP endogenously at comparable levels to our ectopic expression model and produce HA of polydisperse sizes, from 50 kDa to several MDa, in a CEMIP-dependent manner [5]. Skin fibroblasts have similarly been reported to degrade HMW-HA to IMW-HA of 10–100 kDa in size in a CEMIP-dependent manner [4,16]. Overexpression of CEMIP in pancreatic cancer cells also increased the levels of HA < 100 kDa in the extracellular milieu [46]. In contrast, others have reported that CEMIP can degrade HA into fragments of 2–3 disaccharides in length [14]. In this context, we note that we have employed relatively high concentrations of HA in our assays, and thus it is conceivable that IMW-HA produced by CEMIP may act in a concentration-dependent negative feedback manner to suppress further CEMIP-dependent HA degradation. Nevertheless, our results and those from others suggest that HA degradation even with high CEMIP expression levels mainly produces HA in the range of LMW-HA to IMW-HA (10–100 kDa), with the majority of cleavage products being in the IMW-HA range.

While the pro-angiogenic, pro-lymphangiogenic, pro-inflammatory, and pro-migratory roles of HA oligosaccharides and LMW-HA (<10 kDa) are well established [47,48], the biological activity of IMW-HA produced by CEMIP remains poorly investigated, and the available data are often contradictory and inconclusive. Although the pro-angiogenic role of HA seems to be limited to LMW-HA [49], an immunoregulatory activity was shown for various sizes of HA. HA fragments of around 100 kDa in size activate monocytes, increase their cytokine production, and foster monocyte differentiation into immunosuppressive macrophages [50]. Similarly, 200 kDa HA stimulates inflammatory gene expression in macrophages [51], while 500 kDa HA promotes polarization into M2 macrophages [52]. CEMIP-mediated HA degradation has also been linked to cancer cell migration in a size-dependent manner, with 35 kDa HA promoting migration and invasion of cancer cells, while 117 kD HA exerts the opposite effect [46,53]. Similarly, LMW- and HMW-HA inhibit wound closure, whereas 100–300 kDa HA promotes keratinocyte migration**,** increases collagen III expression and enhances wound repair [54,55,56]. Expression of CEMIP during wound healing as we have previously reported [5] would presumably lead to accumulation of HA degradation products of a size that could have a significant impact on these processes. Further research is warranted to investigate the biological and pathological activity of the IMW-HA produced by CEMIP.

Given the role of perturbed HA metabolism in a broad range of pathologies, and the role of CEMIP in numerous diseases (see Introduction), inhibition of the CEMIP hyaluronidase activity has potential therapeutic application. Recently, we identified sulfated HA as a potent CEMIP inhibitor that is active in the low nanomolar range (IC_50_ for sHA3.7 is around 10 nM) [5]. Although flavonoids such as ipriflavone have been proposed as CEMIP hyaluronidase inhibitors [17,36], ipriflavone did not inhibit the CEMIP hyaluronidase activity in our assay, likely because higher concentrations that those we employed here are required to achieve partially significant inhibitory effects [36], suggesting that ipriflavone is a relatively weak CEMIP inhibitor. A number of inhibitors of Hyal1, testicular, streptomyces, and bee hyaluronidases that we tested here, such as N-acetyl-L-cysteine, glutathione, glycyrrhizate, and VcPal, had no significant inhibitory effect on the CEMIP hyaluronidase activity at concentrations of up to 10 µM. It is notable that the IC_50_ concentrations of these compounds for other hyaluronidases were also at least in the same range or even exceeded this concentration, indicating, in general, a limited potency of these substances as hyaluronidase inhibitors [30,33,34,35]. By contrast, in this study, we have identified a number of sulfated polymers as novel and highly potent inhibitors of the CEMIP hyaluronidase activity, notably the naturally occurring sulfated glycosaminoglycan heparin, as well the synthetic substances DXS and PSS.

Taken together, our results suggest that a variety of sulfated hydrocarbon polymers preferentially inhibit CEMIP. Given that sulfated HA inhibits the binding of CEMIP to HMW-HA (Figure 3A), it is likely that competitive inhibition underlies these observations. In future work, we will biochemically characterize the mode of inhibition of these inhibitors in depth. With regard to the inhibitory potency of these substances for CEMIP in comparison to other hyaluronidases, sulfated HA inhibits CEMIP more potently than it inhibits Hyal1 or BTH [5,30]. Heparin inhibits CEMIP and Hyal1 with a similar potency (IC_50_ for CEMIP 0.34 µM (this study), IC_50_ for Hyal1 0.39 µM [30]). For PSS, we found that the CEMIP hyaluronidase activity is around 5–10 times more sensitive to inhibition by this compound compared to Hyal1 or bee venom hyaluronidase [30]. DXS 500 inhibits testicular and Streptomyces hyaluronidases in a concentration range of 100–400 µg/mL (200–800 nM) [31]. Notably, we found that the inhibitory activity of DXS on CEMIP-mediated HA depolymerization is at least two orders of magnitude lower than for these hyaluronidases, indicating that DXS is not only the most potent CEMIP inhibitor discovered to date, but also that it is highly selective for CEMIP compared to the other hyaluronidases tested.

The length of the sulfated polymer and the degree of sulfation clearly impacts on the relative potency of the sulfated polymer to inhibit the CEMIP hyaluronidase activity. The most potent inhibitors were DXS 500 and PSS 1000 with a molecular weight of around 500 or 1000 kDa, and IC_50_ values of 0.5–2 nM. In line with this, our results also suggest that larger sulfated HA derivatives are more potent inhibitors of CEMIP than smaller ones. The size of PSS polymers has also been reported to correlate with their potency to inhibit Hyal1 and testicular hyaluronidase [30]. Moreover, all of these sulfated polymers that inhibit the CEMIP hyaluronidase activity contain more than one sulfation group per disaccharide. In this regard, it is notable that heparin possesses the highest negative charge density of all known macromolecules, and it is interesting to speculate whether sulfation is required for docking into a specific pocket in the CEMIP protein, or whether it is simply the negative charge density of the sulfate groups that is decisive, perhaps through cation sequestration. In this context, it is important to point out, however, that not all sulfated glycosaminoglycans inhibit the CEMIP hyaluronidase activity. Previously, we have reported that chondroitin sulfates do not inhibit the hyaluronidase activity of CEMIP [5]. Although this may be due in part to a relatively low degree of sulfation with only one sulfate residue per disaccharide, sulfated HA with a similar degree of sulfation was able to inhibit CEMIP-mediated HA depolymerization [5]. As chondroitin sulfate is only stereoisomerically different to HA, this strongly suggests that the conformation of the backbone of the sulfated glycosaminoglycan or other sulfated hydrocarbon polymer is decisive for the inhibitory effects on the CEMIP hyaluronidase activity.

Among the sulfated hydrocarbons we describe here as CEMIP inhibitors, heparin is the only one that is naturally produced. It is a specialized heparan sulfate comprising disaccharides of alternating hexuronic acid and hexosamine subunits, and contains an average of 2.7 sulfate groups per disaccharide. Under normal physiological conditions, free heparin is not usually detectable in the blood as it is bound to plasma proteins, and can only be detected after proteolytic digest [57]. However, when heparin is given therapeutically, levels of free heparin of up to 0.5 U/ml can be detected in the blood [58], which corresponds to the median inhibitory concentration of heparin for CEMIP that we determined to be around 1 U /ml (equivalent to 0.3 µM). Functionally, heparin is specifically produced by mast cells and is required for mast cell granule formation [59], and therefore contributes to the regulation of local immune processes. Our results therefore suggest that heparin is likely to contribute physiologically to the resolution of inflammation by inhibiting CEMIP, leading to the accumulation of HMW-HA, increased tissue hydration, and inhibition of mast cell proliferation and cytokine release [60,61].

Polystyrene sulfonates (PSSs) are synthetic polymers made from monomers of the aromatic hydrocarbon styrene. Although PSS has been used to treat hyperkalemia, its efficacy, side effects, and safety were problematic [62]. Our results also show that although PSS is a potent CEMIP inhibitor, cytotoxicity at higher concentrations was observed. This may restrict the potential therapeutic application of PSS for inhibiting the CEMIP hyaluronidase activity.

Dextran is a polysaccharide synthesized by lactic acid bacteria and consists of linear, α-1,6-linked glucose monomers with additional branches from other linkages. DXS is a synthetic, anionic derivative and contains approximately 17–20% sulfur residues corresponding to up to three sulfate groups per glucose monomer [63]. Although DXS is known to induce colonic inflammation upon oral administration and therefore serves as a model for inflammatory bowel disease in experimental animals, it is generally considered to be a safe and bioinert molecule, and it is therefore currently widely used in the material sciences for the development of nanocarriers, for example, as a drug-carrier system for the treatment of cancer and rheumatoid arthritis [64,65]. Given the high expression and contribution of CEMIP to diseases being treated experimentally with DXS-based nanoparticles [2,3,7,12,13], our results suggest that DXS carriers are not as bioinert as is widely assumed and are likely to possess intrinsic therapeutic potential, which should be considered in the experimental design and data interpretation in such studies. This observation is further underscored by the fact that drug-free DXS-containing wafers promote wound healing and corneal re-epithelialization and reduce scarring in a corneal wound model, in contrast to wafers containing unsulfated dextran. While it has been suggested in this context that DXS binds to pro-inflammatory cytokines and exerts anti-fibrotic properties [66], our results suggest that the inhibition of CEMIP-mediated HA degradation might also contribute to these effects of DXS.

Finally, it is striking that the sulfated polymers we identified here as CEMIP inhibitors share not only structural but also functional similarities. Heparin is a well-established anticoagulant, an activity that is probably due to its interaction with antithrombin [67,68]. Both heparin and DXS possess anti-coagulant, anti-inflammatory, and anti-viral properties [64,66,67,68,69]. An anti-inflammatory and a weak anti-coagulant, as well as an anti-viral, activity have also been described for sulfated HA [70,71,72]. The inhibition of the hyaluronidase activity of CEMIP that we show here is an additional common property of these sulfated hydrocarbon polymers. In this regard, it is noteworthy that HA has been suggested to link inflammation and coagulation, for example, in inflammatory bowel disease [73]. Further research is needed to elucidate the molecular mechanisms through which these sulfated hydrocarbon polymers impact on these diverse processes, and thereby to reveal possible functional and mechanistic commonalities.

## 5. Patents

The results from this work were submitted as a patent application under application number EP24215709.

## Figures and Tables

**Figure 2 cells-14-00101-f002:**
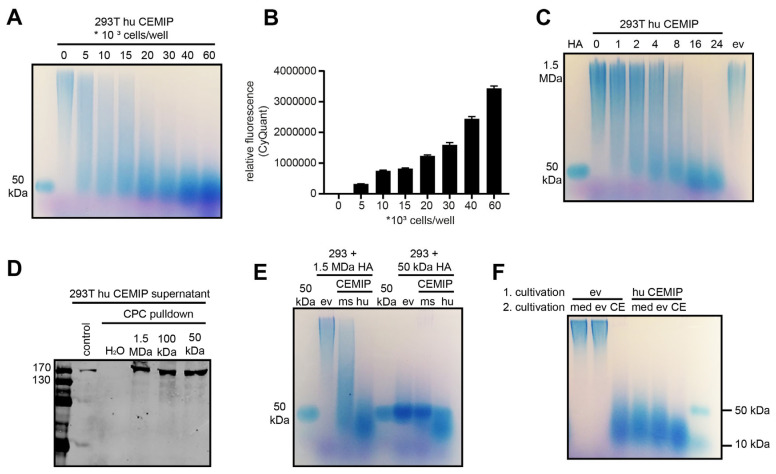
Optimization and analysis of HA degradation by 293T CEMIP. (**A**,**B**) 293T human CEMIP cells were seeded in a 96-well plate in different densities from 5 to 60 × 10^3^ cells/well and cultivated overnight. Cells were then incubated with HMW-HA for a further 24 h, then the medium was harvested. Samples were subsequently analyzed by agarose gel electrophoresis (**A**). A CyQuant assay (**B**) was performed at the end of the experiment to assess cell numbers. (**C**) Time course of HA degradation by human CEMIP. 293T cells overexpressing human CEMIP were seeded in a 96-well plate (40,000 cells/well) and cultivated for 24 h. HA was then added, and aliquots of conditioned medium were harvested after 0, 1, 2, 4, 8, 16 and 24 h. As a control, 293T cells transfected with empty vector (ev) were incubated with HA for 24 h. (**D**) Cetylpyridinium chloride (CPC) pull down of secreted CEMIP with different HA sizes shows that CEMIP binds to HA fragments of 1500, 100 and 50 kDa. The CPC pulldown was performed as described in Materials and Methods. (**E**) 293T cells transfected with empty vector (ev), or mouse (ms) or human (hu) CEMIP were incubated with HMW-HA of 1.5 MDa or HA of 50 kDa for 24 h. The medium was then harvested and analyzed by agarose gel electrophoresis. (**F**) Repeated digestion of HA by CEMIP to determine the size of the minimal degradation products produced by CEMIP. Medium alone (med), 293T empty vector (ev) cells or 293T hu CEMIP (CE) cells were incubated with HMW-HA for 24 h. The HA-containing medium was then collected, diluted 1:1 with fresh medium and added to new, confluent cells seeded the day before, or to medium alone (2. cultivation). After a further 24 h incubation, the conditioned medium was harvested. For all experiments (**A**,**C**,**E**,**F**), the conditioned medium was treated with ProteinaseK, HA precipitated with EtOH and then the same amount of HA was analyzed by 1% (**A**,**C**,**E**) or 2% (**F**) agarose gel electrophoresis. Gels were stained with Stains-all. 50 and <10 kDa HA (Figure (**E**) only) served as size standard.

**Figure 3 cells-14-00101-f003:**
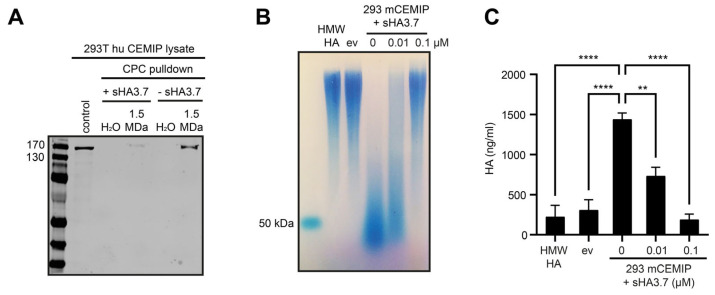
Analysis of CEMIP mediated HA degradation and inhibition by ultrafiltration and HA-ELISA. (**A**) A CPC pulldown with lysates from 293T human CEMIP cells in the presence or absence of 1 µM sHA3.7, added 30 min before HA shows that sHA3.7 prevents binding of HA to CEMIP. (**B,C**) 293T cells transfected with empty vector (ev) or mouse CEMIP were incubated for 24 h with 100 µg/mL HMW-HA. CEMIP-expressing cells were also treated with 0, 0.01 or 0.1 µM sHA3.7 as indicated. The medium was then collected and analyzed in parallel using agarose gel electrophoresis (**B**) and the medium-throughput assay (**C**). (**B**) An aliquot of the conditioned medium was treated with ProteinaseK, HA precipitated with EtOH and then analyzed by 1% agarose gel electrophoresis. (**C**) Triplicate 50 µL samples of the conditioned medium were diluted 1:1 with PBS and centrifuged through 100 kDa ultrafiltration plates. The filtrates were collected and 1:10 dilutions were then analyzed using the ELISA-like HA assay. The mean of technical triplicates +/− SE of dilutions measured in the assay is shown. Significance was calculated using One-way ANOVA between the indicated groups. ** *p*< 0.005; **** *p* < 0.0001.

**Figure 4 cells-14-00101-f004:**
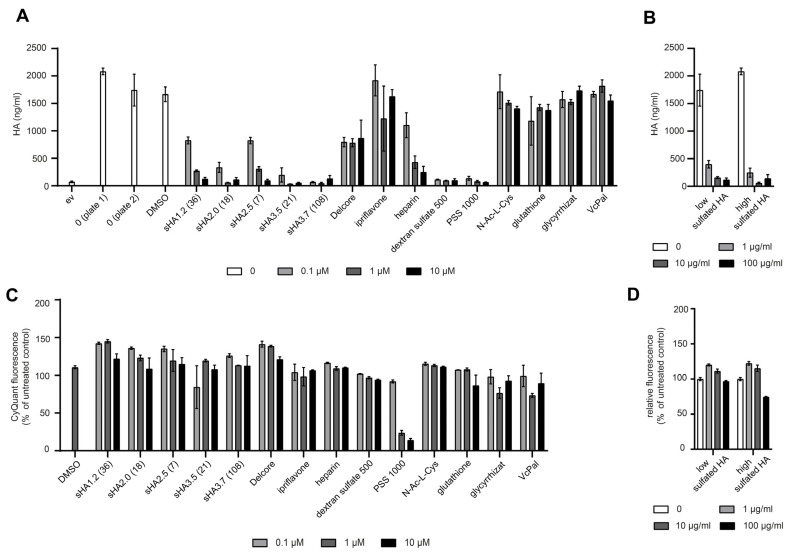
Analysis of potential CEMIP hyaluronidase inhibitors by means of the medium-throughput assay. 293T cells expressing human CEMIP were treated 24 h after seeding with the indicated concentrations of potential inhibitors (for (**A**,**C**): 0.1, 1, 10 µM; for (**B**,**D**): 1, 10, 100 µg/mL). For the different sulfated HA derivatives, the numbers given indicate the sulfation degree and, in brackets, the molecular weight in kDa. After 15 min incubation, 50 µg/mL HMW-HA was added, and the cells were cultivated for a further 24 h. (**A**,**B**) The medium was collected, 50 µL of all samples were then diluted 1:1 and centrifuged through 100 kDa ultrafiltration plates. The filtrates were collected and 1:20 dilutions of the samples were then analyzed by the HA ELISA-like assay. Cells without any inhibitors (analyzed on each plate separately) and cells treated with 0.5% DMSO which was used as solvent for some of the compounds were used as controls. The mean +/− SE (*n* = 3) of the dilutions measured in the assay is shown. In one-way Anova tests, inhibition relative to the respective control was significant for all sHA samples employed (*p* < 0.0001), as well as for Delcore, heparin, PSS and DXS. The inhibitory effect of the other tested compounds was not significant. (**C**,**D**) The number of viable cells was quantified using the CyQUANT Direct assay. The mean +/− SE of triplicate samples normalized to untreated controls is shown (control = 100%). In one-way Anova tests, inhibition relative to the respective control was significant for 1 and 10 µM PSS, as well as for 100 µg/mL high sulfated HA (*p* < 0.0001). The other tested compounds did not have a significant inhibitory effect.

**Figure 5 cells-14-00101-f005:**
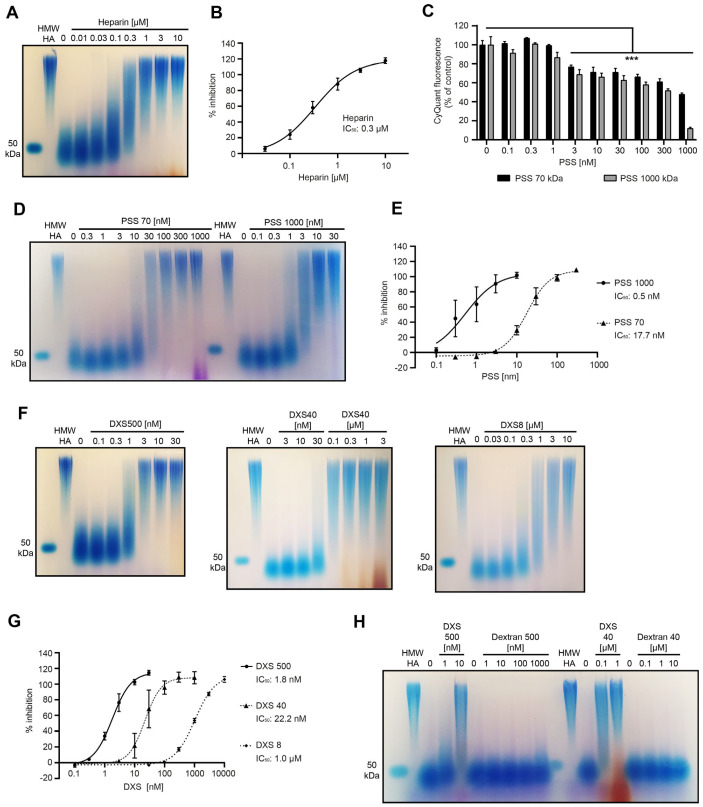
The sulfated hydrocarbons heparin, PSS, and DXS inhibit the hyaluronidase activity of CEMIP. (**A**,**D**,**F**,**H**) HA degradation by 293T human CEMIP-expressing cells in the presence or absence of inhibitor was analyzed by agarose gel electrophoresis and subsequent staining with Stains-All. Cells were treated with the indicated concentrations of heparin (**A**); PSS 70 and PSS 1000 (**D**); DXS 500, DXS 40, and DXS 8 (**F**); and DXS and Dextran (**H**). Please note that concentrations are given in nM or µM. As controls, HWW-HA was incubated in medium without cells, and 50 kDa HA served as a size standard. Each experiment was repeated three times. Representative results are shown. (**B**,**E**,**G**) Quantification of the activity assays is described in the Section 2. For quantifications and IC_50_ calculations, three independent titration experiments were used, and the mean +/− SE is shown in the graphs. (**C**) 293T human CEMIP cells were treated with the indicated concentrations of PSS 70 or PSS 1000 and HMW-HA or were left untreated for 24 h. The number of cells was quantified using CyQuant Direct assays. A representative result is shown. Data represent means +/− SE (*n* = 3). Significant differences between untreated and treated samples were determined using one-way Anova, *** *p* < 0.001.

## Data Availability

The original contributions presented in this study are included in the article/Appendix A. Further inquiries can be directed to the corresponding author.

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
