# Peer review of "A Novel, Cell-Compatible Hyaluronidase Activity Assay Identifies Dextran Sulfates and Other Sulfated Polymeric Hydrocarbons as Potent Inhibitors for CEMIP"

_cells, 2025, doi:10.3390/cells14020101_

Round 1
Reviewer 1 Report
Comments and Suggestions for Authors
This is a well-written and engaging study that focuses on medium-throughput screening to identify inhibitors of CEMIP. The authors utilize ultrafiltration plates to separate low molecular mass HA from high molecular mass HA, followed by quantification using a HA ELISA-like assay. Additionally, they validate the results of the medium-throughput assay for CEMIP-mediated HA degradation against data obtained from agarose gel electrophoresis. Through this assay, the authors identify heparin and dextran sulfate as potent, non-toxic, and novel inhibitors of CEMIP hyaluronidase activity. This study will be highly beneficial for readers and will enhance their understanding of hyaluronan research. I recommend publication after minor revisions.
1. The authors should clarify the Echelon Biosciences HA assay products used, including their catalog numbers. Furthermore, it is important to discuss any experiences regarding the potential unspecific binding of HA to the filters in the 96-well plates. In general, please ensure that catalog numbers and company names are provided.
2. In the “CPC Pulldown” section, the authors must specify the amount of cell-lysed protein utilized.
3. In Fig. 1A and the corresponding text, the authors should clarify whether high molecular weight HA (HMW-HA) is added to CEMIP overexpressing 293T cells, which typically produce minimal hyaluronic acid, or if it is also administered to cell cultures that endogenously express CEMIP and synthesize high levels of HA. Additionally, the rationale for adding HMW-HA (1.5 MDa; 50 µg/ml) needs to be discussed in the Results section.
4. Please provide Intra- and Inter-assay variations
Author Response
This is a well-written and engaging study that focuses on medium-throughput screening to identify inhibitors of CEMIP. The authors utilize ultrafiltration plates to separate low molecular mass HA from high molecular mass HA, followed by quantification using a HA ELISA-like assay. Additionally, they validate the results of the medium-throughput assay for CEMIP-mediated HA degradation against data obtained from agarose gel electrophoresis. Through this assay, the authors identify heparin and dextran sulfate as potent, non-toxic, and novel inhibitors of CEMIP hyaluronidase activity. This study will be highly beneficial for readers and will enhance their understanding of hyaluronan research. I recommend publication after minor revisions.
Comment 1: The authors should clarify the Echelon Biosciences HA assay products used, including their catalog numbers. Furthermore, it is important to discuss any experiences regarding the potential unspecific binding of HA to the filters in the 96-well plates. In general, please ensure that catalog numbers and company names are provided.
Response 1: We thank the reviewer for raising these important issues. We have now included the catalog numbers of these two products in the revised manuscript, and also those of other relevant reagents.
Regarding unspecific binding of HA to the filter of the 96-well plate, we have not investigated directly whether HA remains bound to the filter after centrifugation. However, by using agarose gel electrophoresis to directly compare the amount of HA fragments of less than 50 kDa in the input material with the amount of HA fragments of less than 50 kDa isolated in the filtrate, we know that not all of the HA fragments of less than 50 kDa passes through the filter. Indeed, during assay optimization, the centrifugation conditions in the assay were optimized to maximize the amount of HA fragments of less than 50 kDa that pass through the filter. Importantly, the size distribution of the fragments that pass through the filter is not changed by the ultrafiltration. In addition, quantification of the amount of HA fragments of less than 50 kDa in the medium-throughput assay reflects the amount of HA fragments of less than 50 kDa in the input material, as evidenced by gel electrophoresis (please see Figure 3). This issue has been commented on in the revised text (page 9, last paragraph).
Comment 2. In the “CPC Pulldown” section, the authors must specify the amount of cell-lysed protein utilized.
Response 2: We have included the missing concentration of the protein lysates in the description of the assay (page 5). For the pulldown we generally use 135 µL of protein lysates with concentrations from 0.25 – 1 µg/µL, resulting in 34 – 135 µg protein.
Comment 3. In Fig. 1A and the corresponding text, the authors should clarify whether high molecular weight HA (HMW-HA) is added to CEMIP overexpressing 293T cells, which typically produce minimal hyaluronic acid, or if it is also administered to cell cultures that endogenously express CEMIP and synthesize high levels of HA. Additionally, the rationale for adding HMW-HA (1.5 MDa; 50 µg/ml) needs to be discussed in the Results section.
Response 3: Thank you for pointing out the need for clarification. Indeed, both CEMIP overexpressing 293T cells as well as the fibroblasts that have endogenous CEMIP expression and HA production were treated with 50 – 100 µg/ml exogenously-added HMW-HA. In the revised version of the manuscript, it is now written in the legend for Figure 1 that “all” cells were treated (page 7). Please note that we determined that the endogenously-produced HA levels in the conditioned medium from the MRC5 and MEF cells was around 1 µg/ml under the conditions employed. This means that the majority of HA in Figure 1C results from the exogenously administered HA. We have introduced a comment on this point in the revised manuscript (page 6, last paragraph).
The size of 1.5 MDa HA was chosen as cells usually produce HA in this HMW range (for example Itano et al.,1999; PMID: 10455188). The use of 1.5 MDa HWM HA as a CEMIP substrate facilitated separation of undigested HA from the HA fragments produced by CEMIP hyaluronidase activity using the ultrafiltration plates. The concentration of 50 µg/ml is physiologically relevant (for example synovial fluid HA concentrations reach several mg/ml (Fraser et al., 1997; PMID: 9260563)). Furthermore, this concentration reduces variance in the assay due to any endogenously-produced HA (see comment above). A short rational for adding 1.5 MDa HA at 50 µg/ml is now added in the first paragraph of the result section (page 6)
Comment 4. Please provide Intra- and Inter-assay variations
Response 4: Thank you for suggesting inclusion of these important parameters. Given the multi-component, multi-step nature of the assay we present, calculation of coefficients of variation (CV) for intra-assay variation using a classical technical replicate approach is difficult. The Echelon HA ELISA-like assay, one component of our assay, has by itself an intra-assay CV of 18.9 %, and an inter-assay CV of 9.5 % (Haserodt et al., 2011; PMID: 20864567). We have nevertheless calculated the intra-assay CV for our assay as a whole, and found a strong dependence on the quantity of HA measured. For quantities of CEMIP-generated fragments of HA of 1.3 μg/ml and above in the filtrates, we calculated a CV of 12.7%, based on sixteen triplicate samples. The CV increases progressively when lower amounts of HA fragments are produced by the CEMIP hyaluronidase activity, dropping to 32% for all conditions measured (52 triplicate samples). Similarly, the inter-assay CV was 9% for quantities of HA fragments of 1.3 μg/ml and above, dropping to 54% for samples containing less than 0.2 μg/ml HA. These data reflect that fact that competitive ELISA-like assays (such as the Echelon HA ELISA-like assay that we employed) are very sensitive to variance in samples with low concentrations of analyte, and the fact that cell-based assays generally have a higher CV compared to non-cell-based assays (Li et al., 2024, PMID: 38978387). The high intra- and inter-assay CVs in our assay under conditions in which CEMIP hyaluronidase assay is absent or strongly inhibited – and thus little if any HA fragmentation occurs – means that our CEMIP hyaluronidase assay is not suited for accurate quantification of very low levels of CEMIP activity. This represents a limitation of the assay. A paragraph discussing these issues has now been introduced into the Discussion (pages 15 and 16). Importantly, this limitation does not impact on the application of the CEMIP hyaluronidase activity for screening for inhibitors, as employed in our study.
Reviewer 2 Report
Comments and Suggestions for Authors
In this paper the authors have developed a new method to measure the enzyme activity of the CEMIP hyaluronidase. The hyaluronidase activity of CEMIP was difficult to quantify compared with other hyaluronidases. Thus, this is an important development. In addition, since it is a medium-throughput assay, many compounds can be tested as a CEMIP inhibitor. The authors have found sulfated hydrocarbon polymers including heparin, sulfated hyaluronan, dextran sulfate, and polystyrene sulfonate, are strong inhibitors of CEMIP. The methods they have used are appropriate, and results obtained are shown properly. Data found in this study is very interesting. However, the paper can be improved. After correcting the following points, the paper will be accepted for publication on this journal.
Major points.
1. The type of the inhibition seems to be competitive, based on the structural similarity of hyaluronan and inhibitor polysaccharides. This should be demonstrated by performing additional experiments.
2. The discussion section should be revised. At first it is too long and redundant. It can be reduced into half. General nature of each polysaccharide is not necessary. Although potential clinical application of sulfated polysaccharides is stated in a general way, the concrete example of their application by inhibiting CEMIP is not shown. What case should CEMIP be inhibited? What kinds of diseases the CEMIP inhibitor is applicable for? Such discussion is rather important.
Minor points.
1. Page 2, lines 76-77. TMEM2 has a homology with CEMIP.
2. Page 8, line 359. “ds” is wrong. It is degree of sulfation (page 3, line 148). Hyaluronan is not sulfated.
3. Page 14, line 557. Remove one “been”.
Author Response
In this paper the authors have developed a new method to measure the enzyme activity of the CEMIP hyaluronidase. The hyaluronidase activity of CEMIP was difficult to quantify compared with other hyaluronidases. Thus, this is an important development. In addition, since it is a medium-throughput assay, many compounds can be tested as a CEMIP inhibitor. The authors have found sulfated hydrocarbon polymers including heparin, sulfated hyaluronan, dextran sulfate, and polystyrene sulfonate, are strong inhibitors of CEMIP. The methods they have used are appropriate, and results obtained are shown properly. Data found in this study is very interesting. However, the paper can be improved. After correcting the following points, the paper will be accepted for publication on this journal.
Major points.
Comment 1. The type of the inhibition seems to be competitive, based on the structural similarity of hyaluronan and inhibitor polysaccharides. This should be demonstrated by performing additional experiments.
Response 1. Thank you for this helpful comment. The way in which the sulfated compounds inhibit CEMIP hyaluronidase activity is an important and interesting issue. Please note that in Figure 3A we already show in the manuscript that sulfated HA inhibits the binding of CEMIP to HWM-HA, indicating a competitive inhibition mechanism. We can certainly carry out additional experiments for the other CEMIP inhibitors identified in this study, but would need longer that the 5-day revision period given by the journal to do this. If the Editors consider this to be an important issue in the current manuscript, we can address this issue experimentally but would need to be given more time for the revision. More broadly going forward, a proper biochemical analysis of the mode of inhibition would be helpful, but is beyond the scope of the current paper. In the revised version of the manuscript we have now commented further on the type of inhibition in the Discussion (page 17, second paragraph).
Comment 2. The discussion section should be revised. At first it is too long and redundant. It can be reduced into half. General nature of each polysaccharide is not necessary. Although potential clinical application of sulfated polysaccharides is stated in a general way, the concrete example of their application by inhibiting CEMIP is not shown. What case should CEMIP be inhibited? What kinds of diseases the CEMIP inhibitor is applicable for? Such discussion is rather important.
Response 2. Thank you for this suggestion to help us improve the manuscript. In the revised version, we have carefully edited the Discussion to remove redundancies and to try to reduce the length as far as possible. However, it has not been possible to halve the length of the Discussion without omitting important issues that need to be considered, not least because the reviewers asked us to include some additional points in the Discussion text. Please note that in the Introduction we described the pathologies and diseases in which CEMIP has been implicated - and for which CEMIP inhibition could have application - as this provides the motivation for carrying out this study. Thus, rather than repeating this information in the Discussion, and to comply with reviewer’s request to shorten the Discussion, we refer the reader in the Discussion instead to the Introduction. We hope that the revised version of the Discussion now meets with the reviewer’s approval.
Comment 3. Minor points.
- Page 2, lines 76-77. TMEM2 has a homology with CEMIP.
- Page 8, line 359. “ds” is wrong. It is degree of sulfation (page 3, line 148). Hyaluronan is not sulfated.
- Page 14, line 557. Remove one “been”.
Response 3: Thank you for pointing out these errors. They have been resolved in the revised version of the manuscript
Reviewer 3 Report
Comments and Suggestions for Authors
Authors Schmaus et al., in their manuscript entitled “A novel, cell-compatible hyaluronidase activity assay identifies dextran sulfates and other sulfated polymeric hydrocarbons as potent inhibitors for CEMIP,” present a novel, cell-compatible hyaluronidase activity assay designed to identify potent inhibitors of CEMIP. This assay utilizes ultracentrifugation plates and an HA ELISA-like assay to quantify HA fragments. The study identified several sulfated hydrocarbon polymers, including heparin and dextran sulfate, as potent inhibitors of CEMIP's hyaluronidase activity. Dextran sulfate emerged as the most potent inhibitor with an IC50 of 1.8 nM.
The manuscript brings clear novelty in the form of the presented assay. It is well-structured with interpretations consistent with the obtained data. The following comments should be considered while revising the manuscript.
Major comments:
An interesting question arises based on the time applied for the determination of CEMIP enzymatic activity. Is 96 hours really necessary? Why did the authors not test shorter incubations?
Minor comments:
Page 9, line 392: The effects for sHA3.7 are mentioned; however, the concentrations are not specified here (only later in the discussion around 10 nM, from their previous article). For better clarity, it would be helpful to add it here as well, with a number and citation.
Figure 4 misses the statistical evaluation. The differences are visible, and having it supported by statistics would improve the message. According to Figure 4A, it seems that Delcore has some inhibitory effect on CEMIP. In the text on page 10, line 446, the authors write that "was unable to inhibit." The effect is not dose-dependent and not as strong as, for example, with sHA or dextran sulfate, but it is not completely without effect. It might also be some competitive inhibition of CEMIP when more HA is added, depending on the molecular weight of Delcore.
English Language - typos and grammar:
Is “ultracentrifugation” a correct term for the employed method? It sounds like using a high-speed ultracentrifuge for the analysis.
Page 2, lines 84: "endogeneously" should be "endogenously."
Page 3, lines 140:: "dextan 40" should be "dextran 40."
Page 9, lines 370/371: "Only human but not mouse CEMIP was able to significantly but only marginally degrade HA of 50 kDa." The phrase "significantly but only marginally" sounds strange. Try to rephrase it.
Page 12, line 498: "CEMP-expressing cells" should be "CEMIP-expressing cells."
Page 14, line 557: "been" is written twice.
Author Response
Authors Schmaus et al., in their manuscript entitled “A novel, cell-compatible hyaluronidase activity assay identifies dextran sulfates and other sulfated polymeric hydrocarbons as potent inhibitors for CEMIP,” present a novel, cell-compatible hyaluronidase activity assay designed to identify potent inhibitors of CEMIP. This assay utilizes ultracentrifugation plates and an HA ELISA-like assay to quantify HA fragments. The study identified several sulfated hydrocarbon polymers, including heparin and dextran sulfate, as potent inhibitors of CEMIP's hyaluronidase activity. Dextran sulfate emerged as the most potent inhibitor with an IC50 of 1.8 nM.
The manuscript brings clear novelty in the form of the presented assay. It is well-structured with interpretations consistent with the obtained data. The following comments should be considered while revising the manuscript.
Major comments:
Comment 1: An interesting question arises based on the time applied for the determination of CEMIP enzymatic activity. Is 96 hours really necessary? Why did the authors not test shorter incubations?
Response 1: Please note that the enzymatic activity time in the assay we established was 24 hours. We are not sure why the reviewer mentions a 96-hour period. In initial experiments (Figure 1), the cells were incubated for 72 hours with HMW-HA (24 hours after seeding). After optimization of the assay (Figure 2A, B, C) we then established an incubation time of 24 hours as the standard condition for the assay, to ensure degradation to HA < 100 kDa (Figure 2C). If the formulation in the paper is misleading on this point, we would be grateful if the reviewer could point out exactly where this is in the text so that we can ensure clarity on this point.
Minor comments:
Comment 2. Page 9, line 392: The effects for sHA3.7 are mentioned; however, the concentrations are not specified here (only later in the discussion around 10 nM, from their previous article). For better clarity, it would be helpful to add it here as well, with a number and citation.
Response 2: Thank you for this helpful suggestion. The first line of the first paragraph in Section 3.2 (page 9) now reads as follows: “As a proof of principle for our medium-throughput assay, we used sulfated HA (sHA), a potent inhibitor of CEMIP that we have previously shown has an IC50 for CEMIP of around 10 nM [5]“.
Comment 3. Figure 4 misses the statistical evaluation. The differences are visible, and having it supported by statistics would improve the message. According to Figure 4A, it seems that Delcore has some inhibitory effect on CEMIP. In the text on page 10, line 446, the authors write that "was unable to inhibit." The effect is not dose-dependent and not as strong as, for example, with sHA or dextran sulfate, but it is not completely without effect. It might also be some competitive inhibition of CEMIP when more HA is added, depending on the molecular weight of Delcore.
Response 3: We thank the reviewer for these suggestions. For reasons of clarity, we would prefer not to show the statistical evaluations for all of the data points in the figure, as the large number of bars and stars that would be needed would make the figure unreadable. Instead, we have included a statement in the legend for Figure 4A and B that in one-way Anova tests, inhibition relative to control was significant (p< 0.0001) for all sHA samples employed, as well as for Delcore, heparin, PSS and DXS. The inhibitory effect of the other tested compounds was not significant. We have provided similar statements in the legend regarding the statistical significance of the inhibitory effects observed in the CyQuant assays in Figure 4C and D (page 11 and 12).
The comment about the partial inhibitory effect of Delcore is well taken. However, please note that the apparent inhibitory effect was not dose-dependent. We currently cannot explain this partial non-dose-dependent inhibitory effect. As Delcore is a commercial product, we cannot rule out the possibility of impurities in the preparation that may have had an impact on the assay, particularly as in previous experiments we have not observed an impact of Delcore on CEMIP activity (Schmaus et al 2022). Given the non-dose-dependent partial inhibitory effects of Delcore as well as our previous results, we did not follow up on these observations further. We have modified the formulation of the text to take these observations into account, and to more accurately describe the results obtained (page 11, text before the figure) .
English Language - typos and grammar:
Comment 4. Is “ultracentrifugation” a correct term for the employed method? It sounds like using a high-speed ultracentrifuge for the analysis.
Response 4. Thank you for pointing this out. We of course used ultrafiltation plates only at relatively low-speed centrifugation (1500 g). We have corrected all terms in the revised manuscript.
Comment 5.
Page 2, lines 84: "endogeneously" should be "endogenously."
Page 3, lines 140:: "dextan 40" should be "dextran 40."
Page 9, lines 370/371: "Only human but not mouse CEMIP was able to significantly but only marginally degrade HA of 50 kDa." The phrase "significantly but only marginally" sounds strange. Try to rephrase it.
Page 12, line 498: "CEMP-expressing cells" should be "CEMIP-expressing cells."
Page 14, line 557: "been" is written twice
Response 5. Thank you for pointing out these errors. They have been corrected in the revised text.
Round 2
Reviewer 2 Report
Comments and Suggestions for Authors
In this paper the authors have developed a new method to measure the enzyme activity of the CEMIP hyaluronidase. The hyaluronidase activity of CEMIP was difficult to quantify compared with other hyaluronidases. Thus, this is an important development. Although the authors have found sulfated hyaluronan is a strong inhibitor of CEMIP, the type of inhibition has not been shown. Figure 3A demonstrates only direct interaction between CEMIP and sulfated hyaluronan. Sulfated hyaluronan possibly binds to not only the catalytic site but also to allosteric site. It may inhibit CEMIP in a non-competitive fashion. Additional experiments are required to determine the type of inhibition. Revision period should be extended.
Author Response
Reviewer's comments:
In this paper the authors have developed a new method to measure the enzyme activity of the CEMIP hyaluronidase. The hyaluronidase activity of CEMIP was difficult to quantify compared with other hyaluronidases. Thus, this is an important development. Although the authors have found sulfated hyaluronan is a strong inhibitor of CEMIP, the type of inhibition has not been shown. Figure 3A demonstrates only direct interaction between CEMIP and sulfated hyaluronan. Sulfated hyaluronan possibly binds to not only the catalytic site but also to allosteric site. It may inhibit CEMIP in a non-competitive fashion. Additional experiments are required to determine the type of inhibition. Revision period should be extended.
Response 1:
We thank the reviewer for raising this issue in more detail. Nevertheless, we would respectfully point out that the question raised concerning catalytic and allosteric sites is currently impossible to answer due to the fact that the mechanism through which CEMIP mediates hyaluronan depolymerization remains to be determined. As pointed out in the manuscript “secreted CEMIP has HA depolymerizing activity, but only in the presence of the cells that secrete it”. To be more precise, purified CEMIP or conditioned medium from CEMIP-secreting cells is unable to depolymerize hyaluronan. The hyaluronidase activity is only observed in the presence living cells that actively secrete the protein. This indicates that an additional, as yet unidentified factor(s) is required for CEMIP hyaluronidase activity, in addition to the CEMIP protein itself. One group have reported that the membrane protein ANXA1 is required for CEMIP hyaluronidase activity (Zhang W, Yin G, Zhao H, Ling H, Xie Z, Xiao C, Chen Y, Lin Y, Jiang T, Jin S, Wang J, Yang X. Secreted KIAA1199 promotes the progression of rheumatoid arthritis by mediating hyaluronic acid degradation in an ANXA1-dependent manner. Cell Death Dis. 2021 Jan 20;12(1):102). However, we and others have been unable to confirm these findings. Importantly, the current knowledge base regarding the CEMIP hyaluronidase activity cannot rule out a model in which CEMIP binds to HA (see Figure 3A), but that the HA depolymerizing activity is provided by additional unidentified factor(s) that interact with CEMIP (Domanegg K, Sleeman JP, Schmaus A. CEMIP, a Promising Biomarker That Promotes the Progression and Metastasis of Colorectal and Other Types of Cancer. Cancers 2022, 14, 5093). With this background in mind, we hope that the reviewer can appreciate why the highly relevant question regarding catalytic and allosteric sites cannot currently be answered. The co-authors on the manuscript are currently working together with others to understand the structural biology of the CEMIP hyaluronidase activity with the aim of addressing this issue. These facts underlie the statement in the Discussion “In future work, we will biochemically characterize the mode of inhibition of these inhibitors in depth.”. We hope this this explanation will suffice to address the reviewer’s valid concerns, and that the manuscript can now be published in its current form.